# Generating Attention Maps from Eye-gaze for the Diagnosis of Alzheimer's Disease

**Carlos Antunes**                                            CARLOS.VALDES.ANTUNES@TECNICO.ULISBOA.PT

**Margarida Silveira**                                              MSILVEIRA@ISR.TECNICO.ULISBOA.PT

*Instituto Superior Técnico, Av. Rovisco Pais 1, 1049-001 Lisboa, Portugal*

**Editor:** Editor's name

## Abstract

Convolutional neural networks (CNNs) are currently the best computational methods for the diagnosis of Alzheimer's disease (AD) from neuroimaging. CNNs are able to automatically learn a hierarchy of spatial features, but they are not optimized to incorporate domain knowledge.

In this work we study the generation of attention maps based on a human expert gaze of the brain scans (domain knowledge) to guide the deep model to focus on the more relevant regions for AD diagnosis. Two strategies to generate the maps from eye-gaze were investigated; the use of average class maps and supervising a network to generate the attention maps. These approaches were compared with masking (hard attention) with regions of interest (ROI) and CNNs with traditional attention mechanisms.

For our experiments, we used positron emission tomography (PET) scans from the Alzheimer's Disease Neuroimaging Initiative (ADNI) database. For the task of normal control (NC) vs Alzheimer's (AD), the best performing model was with insertion of regions of interest (ROI), which achieved 95.6% accuracy, 0.4% higher than the baseline CNN.

**Keywords:** Deep learning; Alzheimer's disease; Convolutional neural network; Attention mechanism; Eye tracking; Computer-aided diagnosis.

## 1. Introduction

Alzheimer's Disease (AD) is a chronic brain disorder that accounts for 60% to 80% of dementia cases worldwide (Gaugler et al., 2020) and affects predominantly the elderly.

Symptoms include forgetfulness, difficulty reasoning and mood changes like apathy, wandering, agitation and aggression. The brain presents atrophy due to death of neurons and lower metabolic activity. While there is still no cure for AD, its early detection is crucial, as an effective management of the disease may help prevent the progression to more severe stages. Clinical diagnosis is made by collecting medical and family history, asking relatives about changes in behaviour and conducting mental cognitive tests. Brain imaging, like magnetic resonance imaging (MRI) scans or positron emission tomography (PET) scans has also been recognized as a powerful biomarker, however their interpretation is difficult thus computer-aided diagnosis (CAD) has been requested by clinicians to amplify their diagnostic accuracy (Gauthier et al., 2021).

Currently, the best performing algorithms for AD classification from neuroimaging are convolutional neural networks (CNNs). In these networks, the features are automatically extracted rather than handcrafted, however it is not easy to incorporate medical knowledge.

A recent survey on deep models for medical image analysis concluded that integrating domain knowledge improved the performance of the networks in almost all tasks (Xie et al., 2021). As an example, it states that the attention mechanism is a powerful technique to incorporate domain knowledge of radiologists, because the information about where medical doctors focus helped deep learning models yield better results (Li et al., 2020) (Mitsuhara et al., 2021) (Fang et al., 2019) (Cui et al., 2020) (Xie et al., 2022) (Zhang et al., 2021a). Inspired by these results, in this work we investigate whether the generation of attention maps based on eye-tracking data (physician gaze) can improve the performance of AD diagnosis, by directing the classification model to focus on important regions (determined by domain knowledge). The maps that are obtained are multiplied with CNN feature maps, thus certain locations are highlighted while others are attenuated. Two approaches were investigated for attention map generation. In the first approach, average maps are computed from the doctor's gaze maps. In the second approach, the eye-gaze data is used to supervise a CNN trained to generate attention maps. The inferred maps, like in the first approach, are then multiplied with the feature maps of the CNN that does classification, and whose parameters are trained with the class labels only. Finally, this CNN was also trained with regions of interest (ROI) to compare intuitive domain knowledge with pre-defined relevant regions for classification.

Therefore, the main contributions of this work are:

- Introduction of domain knowledge from eye-gaze data from an expert physician into a state-of-the-art CNN model to perform AD classification.

- Training a deep multiscale network and a U-Net with physician eye-gaze data to predict attention maps.

## 2. Related Work

### 2.1. AD detection models

In the last decade, there have been substantial developments in machine learning classification models for AD detection. CNNs are very effective for AD classification problems and ResNets are by far the most popular type of CNN applied (Korolev et al., 2017; Jin et al., 2019; Ullanat et al., 2021; Liang and Gu, 2021; Zhang et al., 2021c,d; Sun et al., 2021). Nonetheless, some authors used AlexNet (Zheng et al., 2018), Inception (Ding et al., 2019) and VGG (Lee et al., 2021; Turkan and Tek, 2021) or applied an ensemble of methods (Liu et al., 2018). Most studies train models with magnetic resonance imaging (MRI) scans (Korolev et al., 2017; Jin et al., 2019; Ullanat et al., 2021; Liang and Gu, 2021; Zhang et al., 2021c,d; Sun et al., 2021; Turkan and Tek, 2021; Zhang et al., 2021b; Basaia et al., 2019), although still a considerable number use other biomarkers, like PET scans (Zheng et al., 2018; Ding et al., 2019; Lee et al., 2021; Liu et al., 2018; Singh et al., 2017; Lu et al., 2018; Jo et al., 2020; Choi et al., 2020), largely from the Alzheimer's Disease Neuroimaging Initiative (ADNI) clinical datasets.

A recent in-depth study (Khojaste-Sarakhsi et al., 2022) about deep learning applications in AD diagnosis research analyzed about 100 published papers since 2019. Besides identifying many trending technologies, the study recognized the importance of the attention mechanism (AM) and suggested it should be further explored. The idea behind the

attention mechanism comes from human visual attention, which illustrates that human vision typically does not scan the entire scene at once, but rather focuses on selective parts of the whole visual field sequentially, according to the person's needs. The AM therefore can be interpreted as weighted values that represent the importance of each specific part of the image for classification. In CNN models there can be many types of attention, like spatial attention, channel attention, self-attention and layer attention, all of which were employed in the analyzed papers. As for examples of models, Dan J. et al. (Jin et al., 2019) trained a 3D ResNet with one layer of spatial attention (convolution and rectified linear unit (ReLU)), which led to an increase of 2% in accuracy. Ullanant et al. (Ullanat et al., 2021) inserted a residual attention block (Wang et al., 2017) to a vanilla ResNet. Liang S et al. (Liang and Gu, 2021) used one layer of channel attention per stage. Each attention block has global max-pooling for each channel, a convolution with 1x1 kernel, ReLU and dense layers. Zhang Y. et al. (Zhang et al., 2021d) created an attention mechanism inspired by the Squeeze-and-Excitation block (Hu et al., 2018) (channel attention) and got an increase of about 2% in accuracy. Regarding the location of the attention mechanism in the network, most studies place it in the middle of the network or throughout every residual block. However, one author (Zheng et al., 2022) concluded the AM was better placed at the head of the network.

All of the experiments mentioned that used AM were made with MRI scans. No studies that applied attention mechanisms to PET scans were found. Nonetheless, PET scans were chosen for this work, because they can show brain alterations before anatomical changes are observed in MRI scans, which is important for early diagnosis (Mayblyum et al., 2021).

## 2.2. Supervised attention

Since there were no studies on the effect of supervising attention mechanisms with human gaze (domain knowledge) for Alzheimer's disease, we looked at works in other fields.

Yu et al. (2017) showed that spatial attention guided by human eye-tracking data can, in fact, enhance performance, in their case, the performance of generating short text information about brief video clips. They created an AM block that predicts a gaze map per frame of the input video. The inclusion of this AM block improved the results by 3.2% for one language metric.

Li et al. (2020) proposed a CNN for glaucoma detection with an attention mechanism supervised by human attention, called AG-CNN. The human-generated attention maps were used to train the attention prediction subnet of their AG-CNN, which is comprised of a CNN with concatenated features of different layers passed through a deconvolution block at the end. Li's model has considerably better performance than other state-of-the-art methods in his field and increased accuracy by 3.4% when compared to the same model without attention.

Ma et al. (2022) proposed a vision transformer for the diagnosis of breast diseases. They infuse the human expert's prior knowledge to guide the network to focus on the patches with potential pathology. This design leads to higher performance (increased accuracy by almost 1% compared to a standard ResNet50). Moreover, the EG-ViT only introduces the mask operation and an additional residual connection to a vanilla vision transformer. This model has the limitation that it needs to be pre-trained with hundreds of millions of data

samples in order to show better results than CNN. This is especially troublesome for 3D images.

Sheng Wang et al. (Wang et al., 2022) designed a supervised network to assess knee X-ray images for osteoarthritis. This model, called GA-Net, is composed of a ResNet classification network and the supervised attention consistency block. This last component is a CAM visualization/localization module (Zhou et al., 2016). Comparing the ResNet18 with ResNet18+Gaze, the accuracy increased by 2% to 62.8%.

## 3. Data

ADNI is a landmark partnership with the purpose of creating a longitudinal study intended to collect biomarkers of AD. From this database, we retrieved fludeoxyglucose (FDG) PET scans, which show the glucose metabolism in the brain, from participants with baseline and 6, 12 and 24-month follow-ups. 1393 scans from 406 subjects were used, 314 were from AD subjects, 714 were from mild cognitive impairment (MCI) subjects and 365 were normal controls (NC). Table 1 presents demographic and clinical information of the study subjects. All FDG-PET had been normalized, averaged and co-registered by ADNI, and were also further normalized to the [0,1] range.

Table 1: Clinical profile of the subjects in three categories (AD, MCI, NC) categories. Age and MMSE are average values with the standard deviation in parenthesis. MMSE refers to the Mini-Mental State Exam, a mental cognitive status assessment that evaluates memory, thinking and simple problem-solving abilities, where the maximum (best) score is 30.

| Group | AD | MCI | NC | All |
|---|---|---|---|---|
| No. of subjects | 95 | 207 | 104 | 406 |
| Age | 76.6 (7.1) | 76.0 (7.3) | 77.0 (4.8) | 76.4 (6.7) |
| Sex (% M) | 59.9 | 65.8 | 63.8 | 64.1 |
| MMSE | 21.1 (4.1) | 26.7 (2.8) | 29.1 (1.2) | 26.1 (2.9) |

Additionally, several PET scan images in this dataset have been complemented with records of the gaze of a medical doctor while performing a diagnosis, thus collecting areas of interest (domain knowledge). This was performed by Bicacro et al. (Bicacro et al., 2012), using a Tobii™ device. For their study, the gaze (a total of 4261 fixation points) for scans of 177 subjects (59 of each category - AD, MCI, NC) was collected. Table 2 presents the proportion of each type of scan within the overall dataset. It is noteworthy that the amount of scans with fixations is only 12.6% of the total scans available. Even though these eye-gaze data have been applied before in (Bicacro et al., 2012) and (Morgado et al., 2012), it was never employed in deep learning models. They were used for selecting and extracting features that were then fed to a support vector machine classifier.

For each scan, the eye-tracker provides discrete fixation points. However, the physician does not look at a particular pixel, but instead looks at a region centered in the fixation

Table 2: PET scans in various categories. Several different scans correspond to the same patients in different periods of the ADNI's longitudinal study, therefore there are more scans than subjects.

| Group | AD | MCI | NC | All |
|---|---|---|---|---|
| **No. of scans** | 314 | 714 | 365 | 1393 |
| **Proportion of total (%)** | 22.5 | 51.2 | 26.3 | 100 |
| **Proportion of scans with fixations (%)** | 4.2 | 4.2 | 4.2 | 12.6 |

point and symmetrically spread out by the visual angle. Therefore, we convolve the fixation map $f(x)$ (image with the points where the doctor focused) with an isotropic bi-dimensional Gaussian function $G_\sigma(x)$, creating an attention map $S(x)$, like in Figure 1 ($(a)$, $(b)$, $(c)$) (image where the regions people's eyes focus are highlighted). The circular region is modeled by the isotropic Gaussian filter and the visual angle by the standard deviation ($\sigma = 3$). Some examples of the resulting maps are shown in Figure 1, where average maps are also shown ($(d)$, $(e)$, $(f)$), given the variability in attention maps.

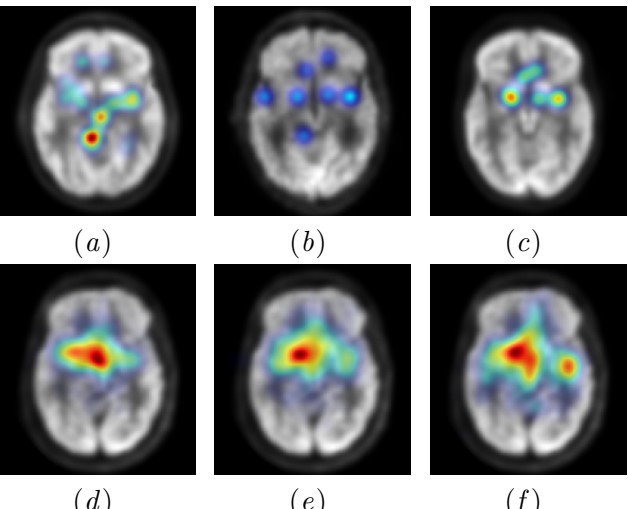

$(a)$ $(b)$ $(c)$

$(d)$ $(e)$ $(f)$

Figure 1: Examples of axial cut 25. The first row shows attention maps obtained by Gaussian filtering of the fixation points for three random patients, with NC ($a$), MCI ($b$) and AD ($c$). The second row shows average attention maps for NC ($d$), MCI ($e$) and AD ($f$).

The same expert physician has manually identified 12 regions of interest (ROI), as displayed in Figure 2. These regions include the lateral and mesial temporal, inferior frontal gyrus/orbitofrontal, inferior and superior anterior cingulate, dorsolateral parietal, posterior cingulate, and precuneus. These anatomical regions of the brain are considered by the

doctor to be the most relevant for the task of AD diagnosis. If we compare the regions of interest with the regions where the doctor looked at, we discover that only 36.2% of fixations fall inside the ROI. This might be concerning since it seems there is little coherence between the regions identified by the doctor and the regions where he focuses his gaze.

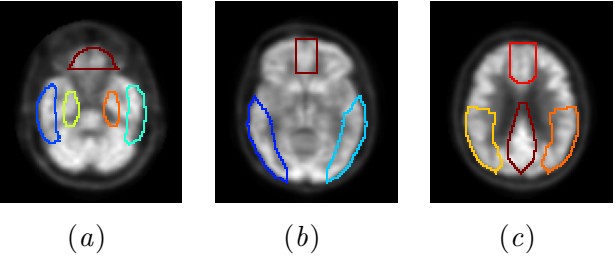

(a)  (b)  (c)

Figure 2: Examples of three axial slices with regions of interest (ROI) defined by the expert physician. (a) Red - Inferior frontal gyrus/Orbitofrontal; Dark and light blue - Lateral temporal; Light green and orange - Mesial temporal; (b) Dark and light blue - Lateral temporal; Red - Inferior and superiro anterior cingulate ; (c) Light red - Inferior and superior anterior cingulate ; Yellow and orange - Dorsolateral parietal; Dark red - Posterior cingulate and precuneus. Some slices do not contain any anatomical ROI.

## 4. Method

In this section, the different models studied are detailed. First, we present the two models investigated for attention mechanism supervision, then we present our approaches that use constant attention maps, either based on average eye-gaze data or from ROIs. Finally, we present our baselines which include a standard ResNet18 and the ResNet18 with attention mechanisms (either CBAM or Residual Attention).

### 4.1. Supervised attention mechanism

In this method, the model is composed of two sub-networks. The first network is used to predict the attention maps, and is supervised by the doctor's fixation maps. The second network is a standard ResNet18, where the created attention mechanism maps are inserted. Two alternatives for generating the attention maps from the doctors' eye-gaze were investigated. The first alternative is the deep multiscale network (Figure 3), which is similar to the glaucoma paper's (Li et al., 2020) attention prediction subnet, but adapted for 3D images and with resizing performed with average pooling and upsampling instead of bilinear interpolation. The encoder portion is a typical CNN, where the input passes through several residual blocks to extract hierarchical features. The decoder portion takes features from distinct basic blocks, normalizes them to the same dimensions, and concatenates them to perform convolutions four times, before applying convolution transpose twice.

The second alternative is a U-Net (Figure 4), which is also an encoder-decoder network. The encoder part performs feature extraction and learns abstract representations of the

input image with convolutions. Here, the spatial dimensions decrease with max pooling operations. Furthermore, the network has two skip connections between the encoder and decoder part, that concatenates two arrays, to be used in the next decoder stage. This helps to provide additional information to the decoder and assists in the flow of the gradient while backpropagating, since it is a shortcut. The decoder section takes the representations to generate the mask. It increases the size through upsampling.

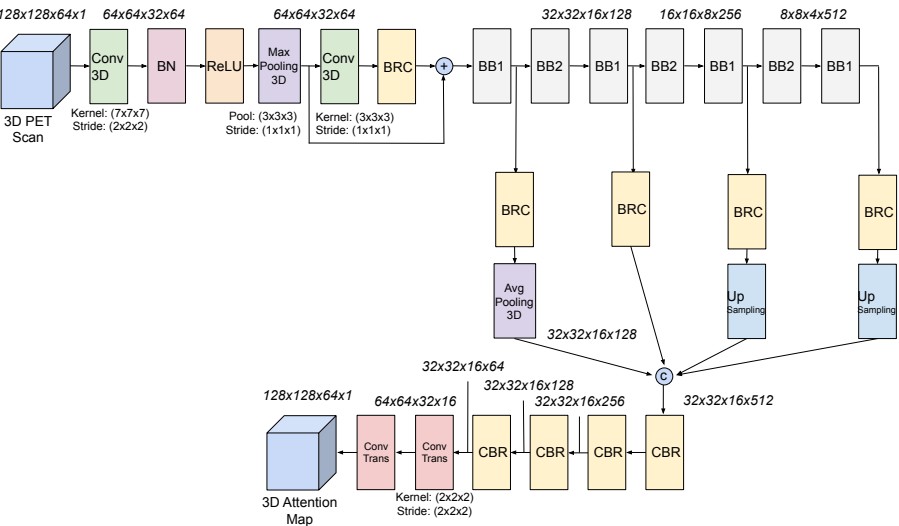

Figure 3: Representation of deep multiscale network that was chosen to learn the attention maps. BRC means batch normalization, ReLU and convolution layers.

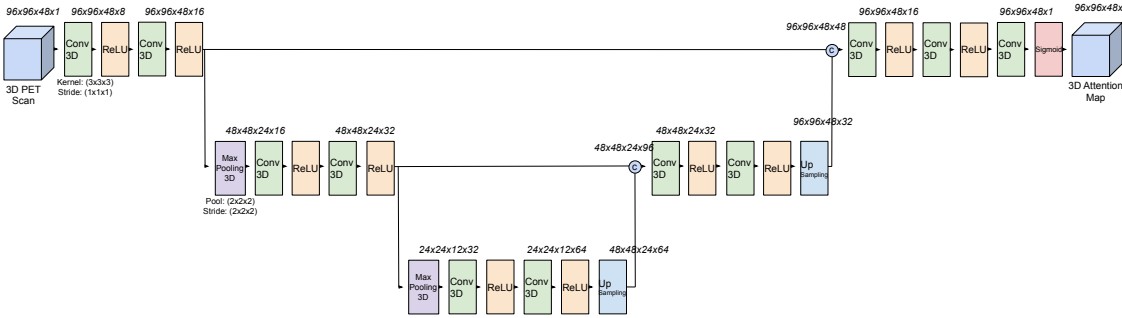

Figure 4: Representation of a 3D U-Net, the second network investigated for AM generation with eye-gaze supervision.

## 4.2. Constant average maps and ROI

In this approach, the attention maps are not created by layers with learned weights. Instead, the doctor's constant average attention map (based on the eye-tracking data) and the ROI maps (hard attention) are introduced into the network, without learning. These maps are inserted in the ResNet18 in the same place as the CBAM module.

## 4.3. Baseline CNNs

The simplest baseline is a vanilla 3D ResNet18. This is an appropriate model since residual networks are considered state-of-the-art and have been widely applied for AD classification. In fact, 38% of the 74 papers that used CNNs for AD diagnosis analyzed by Khojaste-Sarakhsi et al. used ResNets (Khojaste-Sarakhsi et al., 2022). Although this network does not include attention, we can visualize the regions of the input scans that the model considers more important with guided back-propagation (Springenberg et al., 2014) or Grad-CAM (Selvaraju et al., 2017).

Two additional baselines were tested, which integrated attention mechanisms into the ResNet, but that do not incorporate domain knowledge. One attention mechanism is CBAM (Woo et al., 2018), a commonly used attention module that can be integrated into any CNN.

CBAM sequentially infers attention maps along two separate dimensions, channel and spatial, which are multiplied by the input of the respective layer creating a refined feature map. For this study, CBAM was adapted for three dimensions, the same as the scans. To better understand the importance of the spatial attention component, the experiments were also done with the spatial attention sub-module only. The CBAM block was inserted in three different locations (one per trial): at the start of the network before any operation, in the middle basic block, and throughout the basic blocks of the ResNet.

Another attention mechanism tested is residual attention (Wang et al., 2017). This is another type of spatial and channel attention. It uses a bottom-up top-down structure to learn the mask. It collects global information and later guides input features in each position.

## 4.4. Experimental setup

The baseline CNN, the ResNets with CBAM and residual attention and the networks with constant maps/ROI were trained with categorical cross-entropy as the loss function, which was minimized with stochastic gradient descent optimizer for a maximum of 50 epochs. The learning rate was $1 \times 10^{-2}$. Train and testing were done using stratified 5-fold cross-validation. Since we have multiple scans of the same subject at different times, the subjects, and not the images, were separated into five folds. This methodology guarantees that brain scans from the same subject are not present in different sets, thus avoiding data leakage. About 15% of the available samples for training in each fold were used for validation. The model of the epoch with the lowest validation loss was selected as the best model to be tested. The supervised attention mechanism networks (deep multiscale network and U-Net) were trained like the aforementioned models but with Dice coefficient as loss. All models were created with the keras/Tensorflow package on Google Colab notebooks. The main components can be found in this link: https://tinyurl.com/GitHubPaperCode. The classification tasks performed were NC vs AD and NC vs MCI vs AD.

## 5. Results and discussion

The results (accuracy, sensitivity, specificity and $F_1$-score) for the task NC vs AD and NC vs MCI vs AD are displayed in Table 3 and Table 4, respectively. All the models include a ResNet18. The tables only show the results for the best location of the attention mechanism (start, middle or throughout the network), as specified in the 'AM Location' column. The statistical significance of the differences between the results of each AM strategy and the baseline Resnet were evaluated with paired t-tests.

Table 3: Results of 5 fold cross validation for the task NC vs AD. Format: Mean (standard deviation), best result in bold. The lower section consists of models with domain knowledge, while the upper section does not. All the models are composed of a ResNet18 and the AM module specified in the first column.

| Models | AM Location | ACC (%) | SEN (%) | SPE (%) | $F_1$-score (%) |
|---|---|---|---|---|---|
| Standard ResNet18 | - | 95.2 (1.7) | 95.0 (2.4) | 95.3 (2.4) | 94.8 (1.9) |
| CBAM | middle | 94.9 (2.0) | 94.7 (2.7) | 95.3 (2.7) | 94.6 (2.4) |
| CBAM spatial module | middle | 95.0 (1.7) | 94.8 (3.1) | 95.2 (3.4) | 94.6 (2.4) |
| Residual attention | throughout | 95.5 (2.2) | 94.7 (3.8) | **96.2 (2.3)** | 94.8 (2.4) |
| Constant average map | middle | 94.8 (1.5) | 93.7 (3.4) | 95.9 (2.4) | 94.4 (1.5) |
| ROI | start | **95.6 (2.6)** | **95.1 (2.5)** | 96.1 (2.8) | **95.2 (2.7)** |
| Deep multiscale network | start | 94.0 (1.9) | 92.9 (4.0) | 94.9 (2.4) | 93.4 (2.8) |
| U-Net | middle | 95.2 (2.1) | 94.7 (4.0) | 95.6 (2.2) | 94.6 (2.1) |

For NC vs AD, the model with the highest accuracy was ResNet18 with ROI inserted in the start, achieving 95.6% accuracy. This was a 0.4% rise compared to the standard ResNet18, which is statistically significant (p-value<0.05), and the best performing model with domain knowledge.

Figure 5 displays a brain scan overlapped with heatmaps generated by guided backpropagation $(a)$ and Grad-CAM $(b)$ techniques of the standard ResNet18, as well as a scan with fixation points and ROI $(c)$ for comparison. The red areas mean these regions are more important for the classification task. The most important regions for the guided backpropagation mode are slightly different than the ones activated by the Grad-CAM method, except for the center of the brain, which has some red regions for both types of images. The Grad-CAM maps are more similar to the doctor fixations than to the ROI. Nonetheless, from these types of images, no indisputable pattern stands out as a determinate location of the disease.

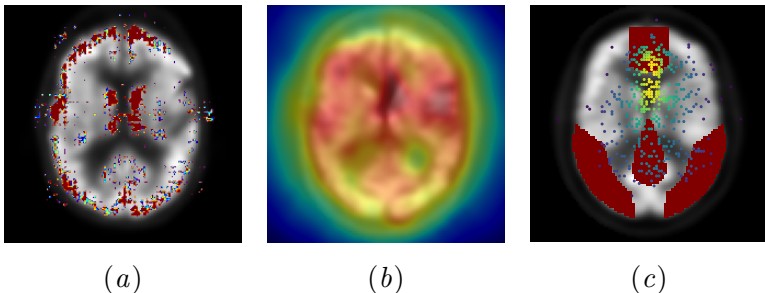

$(a)$              $(b)$              $(c)$

Figure 5: Examples of localization maps using Guided Backpropagation $(a)$ and Grad-CAM $(b)$ techniques for the standard ResNet18 for the task NC vs AD. Each of the maps presented is an average of the generated guided backpropagation and Grad-CAM output for all AD scans available. These maps highlight the regions of the brain image more activated in the network to make the prediction. $(c)$ ROI and fixation points are displayed for comparison.

Examples of the generated attention maps are presented in Figure 6. We computed the Pearson correlation between these maps and the original fixation maps (results not shown) and concluded that the deep multiscale net created maps more similar to the original than the U-Net. Despite this, the U-net obtained slightly better performance and was the best method that incorporated the doctor's attention. Nevertheless, it was not able to obtain better performance than the baselines (p-values<0.05). Some reasons can be hypothesized: the eye-gaze dataset was too small, specially for deep learning which needs a lot of data; the methods of incorporating the eye-gaze were not the most suitable (other approaches were suggested, for example, a supervised CAM module (Wang et al., 2022) or a vision transformer with domain data (Ma et al., 2022)); the assumption that the doctor relies only on the intensity of the voxels to make decisions may be very simplistic, perhaps the doctor is comparing different regions' average intensity, performing basic computations or the mental process of information is different according to the region being analyzed.

For the task NC vs MCI vs AD, the best performing model is the ResNet18 with a constant average duration map in the middle, with 87.4% accuracy (+0.3% than standard ResNet18 and p-value<0.05). This means a different conclusion than for the task NC vs AD, for which the best performing model was with ROI. Therefore, perhaps the ROIs are optimized for AD regions and do not take into account MCI, while the eye-gaze was retrieved when the doctor was performing a classification task that included MCI (NC vs MCI vs AD), thus the constant average maps include this information.

The accuracy results of incorporating the CBAM spatial model and residual attention were not statistically different from those in the baseline ResNet for the binary task, but were statistically significant for the ternary task.

Figure 7 shows the accuracy of our models (in green) juxtaposed with the state-of-the-art networks for better comparison (in gray and blue). This figure shows that our deep models outperformed many of the studies found in the literature. Yet, these comparisons need to be taken lightly because different models were trained, with different biomarkers

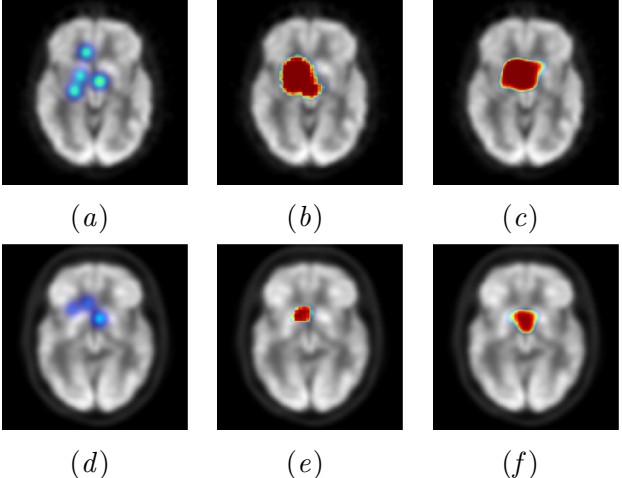

Figure 6: Examples of generated attention maps for axial cut 25 (first row - AD; second row - MCI). The doctor fixation maps are on the left, the deep multiscale network generated attention maps are in the middle and the U-Net maps on the right.

Table 4: Results of 5 fold cross validation for the task NC vs MCI vs AD. Format: Mean (standard deviation), best result in bold. All the models are composed of a ResNet18 and the AM module specified in the first column.

| Models | AM Location | ACC (%) | $F_1$-score (%) |
|---|---|---|---|
| Standard ResNet18 | - | 87.1 (1.6) | 86.7 (1.8) |
| CBAM | middle | 85.9 (1.6) | 85.3 (1.5) |
| CBAM spatial module | middle | 86.9 (2.0) | 86.5 (2.1) |
| Residual attention | throughout | 85.5 (0.9) | 84.5 (3.8) |
| Constant average map | middle | **87.4 (1.5)** | **86.9 (1.1)** |
| ROI | start | 86.2 (1.4) | 85.9 (1.7) |
| Deep multiscale network | start | 86.6 (2.3) | 86.3 (2.0) |
| U-Net | middle | 86.0 (2.3) | 85.5 (2.1) |

and with a different number of scans. The figure also highlights that incorporating domain knowledge helped increase accuracy with ROI for the binary task and constant average maps for the multiclass task.

Our methods also performed better than most expert physicians in NC vs AD classification, who correctly predict 85.7% of scans on average (Klöppel et al., 2008).

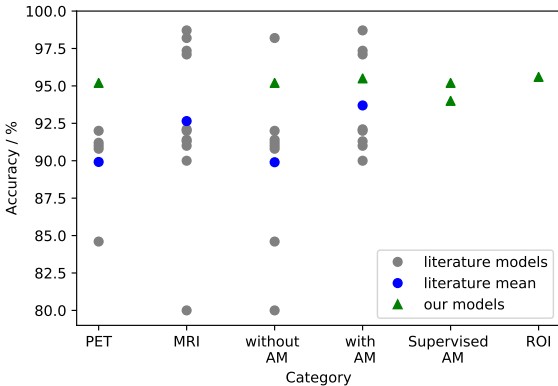

Figure 7: Comparison with state-of-the-art, for the task NC vs AD. Accuracy of our models (in green) contrasted with the models reviewed (in gray) and with the average value of the state-of-the-art models (in blue). 'PET' denotes models trained with PET scans only; 'MRI' denotes models trained with MRI scans only; 'without AM' denotes models without attention mechanism; 'with AM' denotes models that use attention mechanism; 'Supervised AM' refers to our models with supervised attention mechanism (there are no other state-of-the-art models in this category). Finally, on the right, the model with ROI, which was the best performing model.

## 6. Conclusion

In this work we investigated methods to integrate physician attention patterns obtained from eye-tracking data into CNNs for Alzheimer's Disease diagnosis. We explored the use of average gaze-maps and the supervision of a CNN to predict attention maps. We also compared these approaches with the use of ROI hard attention maps.

Our methods performed better than most CAD systems for AD working with FDG-PET images found in the literature. The ResNet18 with the ROI yielded the best results for NC vs AD, with an accuracy of 95.6% and the ResNet18 with constant average maps (Gaussian filtered eye gaze) achieved 87.4% for NC vs MCI vs AD task. These outcomes motivate further work like the creation of a bigger dataset, with more gaze data, following other approaches of introducing domain knowledge, like the visual transformer (Ma et al., 2022) or a CAM module (Wang et al., 2022) and extracting more information from the data besides just the voxel intensity of the "looked at" regions.

## Acknowledgments

This work was supported by LARSyS - FCT Project UIDB/50009/2020.

The PET scans and subjects' data used in the preparation of this article were obtained from the ADNI database (https://adni.loni.usc.edu/). As such, the investigators within the ADNI contributed to the design and implementation of ADNI and/or provided data but did not participate in the analysis or writing of this report.

The source of the highly experienced medical input (eye gaze and ROI) was Dr. Durval Campos Costa, a nuclear medicine expert, from the Champalimaud Foundation. While the method of acquisition and treatment of the eye-tracking data was performed by Eduardo Bicacro at Instituto Superior Técnico.

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
