# OpenReview forum: "Generating Attention Maps from Eye-gaze for the Diagnosis of Alzheimer's Disease"
_NeurIPS.cc/2022/Workshop/GMML — Gaze Meets ML 2022 Poster_

### Official Review · Reviewer_wMgu · 2022-10-13
**Review on Generating Attention Maps from Eye-gaze for the Diagnosis of Alzheimer's Disease**

**Rating:** 6
**Confidence:** 4

**Review:**

## Paper Summary
The authors try to improve the classification performance of deep learning models for Alzheimers disease diagnosis. The approach is to include domain knowledge by physicians using eye tracking data while PET-image based diagnosis is done. The eye tracking data is used to create attention maps which help to emphasize areas of interest and attenuate the irrelevant. The paper relies on PET scans since those show brain alterations early and thus are better suited for early detection of AD.

## Quality
The models are well chosen based on literature. The related work is comprehensive, yet well summarized and presented. A potential research gap is worked out and defined.

In the Results part "paired t-test Wilcoxon tests" are used. I believe this is wrong, since t-test is parametric and Wilcoxon is non-parametric. Thus, a combination of both seems irrational. Which test was used? Did you test for normality? Please clarify that. Furthermore, as far as I understood, the test significance is based on five accuracy values for each group based on the 5-fold CV. The mean (std) of those five values is presented in the tables. It seems unrealistic that those small differences create significant results. To better understand and also for reproducibility of the results you may include an appendix listing the accuracy values for each fold. In case the p-values indicate significance, please also calculate an effect size, since the significance test is only based on a very small group.

The conclusions drawn from Figure 7 "This figure shows that our deep models outperformed the studies found in the literature." is optimistic. The Figure shows that the authors' models outperforms other PET-based models. However, with and without AM models are not outperformed. The authors' model are above the mean, still models are clearly above the authors' performance. Thus, the aforementioned statement has to be rejected.

The manuscript needs proof reading and spell checking. Two examples:
Line 49: ... multiscale network network and ... (network doubled)
Line 88: ... input video. the inclusion ... (missing capitalization)

The formatting of references is odd. I assume this is due to referencing in latex using multiple "\cite{}" after each other. I encourage to comma separate the references within one single \cite{}. If this is not the case, it seems to be a template issue and the authors might ignore this comment.

## Clarity
The paper is well structured and written. Thus, easy to follow.
The methods part, however, lacks clarity in the description of the AM Location for the supervised attention mechanism models and the constant average maps and ROI models. In section 4.3 the CBAM block location is directly addressed. I encourage to restructure the whole methods part. A possible solution would be to explicitly name the AM locations in the experimental setup instead of the individual models' description.

## Originality
The work seems novel and not researched yet.

## Significance
The work is significant, although the current results do not significantly add to the current state of the art.
Still, the approach should be investigated further by collecting more data.

## Final Statement
The work is interesting and worth publishing, however in the current state there are too many weaknesses.
Thus, the rating is marginally above acceptance threshold.
If the comments are addressed, I highly recommend for an accept.

---

### Official Review · Reviewer_yzNn · 2022-10-14
**Review for "Generating Attention Maps from Eye-gaze for the Diagnosis of Alzheimer's Disease"**

**Rating:** 4
**Confidence:** 3

**Review:**

SUMMARY: This paper investigates the use of attentional maps generated from expert eye-gaze information as a prior for deep models designed for AD diagnosis from neuro-imaging data, i.e. Positron Emission Tomography (PET) scans. Their proposed framework modifies standard CNN architectures to incorporate attention generated from eye-gaze and compares this approach against (1) vanilla architectures (2) traditional data-driven attention and (3) (pre-specified) ROI based feature selection.

Evaluation is performed on the Alzheimer’s Disease Neuroimaging Initiative (ADNI) database for two tasks - binary normals vs Alzhiemer's Disease (AD) classification and three class AD vs Mild Cognitive Impairment (MCI) vs healthy controls discrimination. The results suggest that the best performance is obtained from ROI based methods.

STRENGHTS: The paper is straightforward to read and understand, and the presentation of the methodology is clear. Implementation details are transparent and code has been provided for reproducibility.

WEAKNESSES:

1. The attention mechanisms investigated to incorporate eye-gaze (UNet Predictions, Deep Multi-scale Networks) read as off-the-shelf adaptations of standard models. Thus, the paper does not appear to make a significant methodological advance in the workshop subject area.

2. Performance Comparison: From the results of Table 2, the performance for several models is quite similar (differences in the first decimal point). It is unclear from the table which comparisons reach statistical significance thresholds.

 In text, the authors indicate that "This was a 0.4% rise compared to the standard ResNet18, which is statistically significant (p-value<0.05), and the best performing model with domain knowledge." Perhaps, the same testing procedure could be used for comparing across attentional baselines as well. It would also be good to know what statistical test was used for comparison.

3. Section 4.2 "In this approach, the attention maps are not created by layers with learned weights. Instead, the doctor’s constant average attention map for each class (based on the eye-tracking data) and the ROI hard attention) are introduced into the network, without learning." Since the average maps are defined class-wise during training, how are the average maps computed for testing data?

4. It is a bit unclear to me what the intended takeaway of the paper is, given all the caveats mentioned in the discussion. Specifically
 (a) The performance profile for the three-way classification (Table 4) task is very different from that of the AD vs NC task, one suggesting that ROI based methods are better than gaze informed supervision and the other suggesting the opposite
 (b) Only one dataset was used for evaluation. The gaze data was available for only a small percentage of patients.
 (c) Evaluation was performed only against a small subset of standard baseline models (ResNet 18 backbone)

OVERALL RECOMMENDATION: Given all the points raised under weaknesses, I am not sure this paper is mature enough for publication at this stage.

---

### Meta-Review · Area_Chair_gY9z · 2022-10-20

**Recommendation:** Accept (Poster)
**Confidence:** 5

**Metareview:**

The authors present a methodology on integrating eye tracking to improve accuracy of CNNs for detection of Alzheimer's disease on PET images. This is an interesting direction that is being explored in other areas of radiology imaging. While the results of the proposed methodology are borderline improvement to baselines it is worth publication due to the potential applications in radiology imaging.

---

### Decision · Program_Chairs · 2022-10-20

Accept (Poster)